# Aging and Bimanual Effects on Finger Center of Pressure during Precision Grip: Different Strategies for Spatial Stability

**DOI:** 10.3390/s21248396

**Published:** 2021-12-16

**Authors:** Ryoto Akiyama, Naoto Noguchi, Ken Kondo, Koji Tanaka, Bumsuk Lee

**Affiliations:** 1Division of Rehabilitation Service, Geriatrics Research Institute and Hospital, 3-26-8, Odomo, Maebashi 371-0847, Japan; akiyamaryoto@gmail.com; 2Graduate School of Health Sciences, Gunma University, 3-39-22, Showa, Maebashi 371-8514, Japan; noguchinaoto@gunma-u.ac.jp (N.N.); kojit929@gunma-u.ac.jp (K.T.); 3Department of Occupational Therapy, Faculty of Rehabilitation, Gunma Paz University, 1-7-1, Tonya, Takasaki 371-0006, Japan; kenkondoot@gmail.com

**Keywords:** center of pressure, spatial stability, precision grip, bimanual interference, elderly, sensor sheet

## Abstract

The purpose of this study was to examine aging and bimanual effects on finger spatial stability during precision grip. Twenty-one older and 21 younger adults performed precision grip tasks consisting of a single task (grip and lift an object with the thumb and index finger) and a dual task (the grip-lifting task with one hand and a peg board task with the other hand). The center of pressure (COP) trajectory and the grip force were evaluated using a pressure sensor with a high spatial resolution. In the COP trajectory, the main effects of age for the thumb (*F*_1,140_ = 46.17, *p* < 0.01) and index finger (*F*_1,140_ = 22.14, *p* < 0.01) and task difficulty for the thumb (*F*_1,140_ = 6.47, *p* = 0.01) were significant based on ANCOVA. The COP trajectory was statistically decreased in the older adults. The COP trajectory was also decreased in the dual task, regardless of age. The results suggest the existence of a safety strategy to prioritize the spatial stability in the elderly group and in the dual task. This study provides new insights into the interpretation of the COP trajectory.

## 1. Introduction

Age-related changes of physiological and anatomical parameters lead to motor deficits in upper-limb motor control [1,2,3]. In particular, impaired control of bilateral arm movements in older adults could have a negative impact on their daily activities [4,5]. The decreased performance in the simultaneous use of two hands is known as bimanual interference. It is known that bimanual interference is mainly related to task difficulty or dual task demands [6]. For example, compared to the movements that each hand makes towards a common goal (e.g., one hand opening a drawer and the other hand taking out an object), a task that each hand makes based on an independent goal (e.g., opening different drawers simultaneously with each hand) is more complicated. As the task goals are incompatible with each other, interference is more likely to occur [7,8,9]. Considering the fact that most bimanual activities in our daily lives require different actions for each hand [10,11], it is important to understand bimanual interference during object manipulation in the elderly.

Grip force (GF) is the standard parameter for analyzing the precision grip underlying object manipulation. Numerous studies have identified that minimum GF is used to optimize sensory information and conserve energy during unimanual precision grip [12,13]. In contrast, studies of bimanual interference on precision grip are limited. One study investigated the interference effects of lifting/tapping/typing with one hand on contralateral GF control and found that increased GF was a consequence of preparation/instruction for a complex task involving bimanual manipulation [10]. The results suggest that the force control could be affected in situations requiring different actions for each hand. However, considering that combining both the GF direction and magnitude is required to stably hold a freely moveable object [14], measuring only the GF amplitude is not enough to accurately evaluate kinetic changes during precision grip.

The center of pressure (COP) may contribute to solving this problem. The COP is the central point of all applied forces, and it indicates the balance of the forces on the surface between an object and the fingertip(s). It is known that displacement of the COP is induced by deviated finger force direction [15]. Based on this biomechanical relationship, the COP evaluates the finger spatial stability during object manipulation from different perspectives to the GF. Actually, although there was no difference in the GF amplitude, the COP trajectory was increased in the paretic finger compared to the non-paretic one in stroke patients [16]. Moreover, digital anesthesia, median nerve block, polysensory neuropathy, or local muscle fatigue increased the COP displacement during precision grip [17,18,19,20]. These findings imply that the COP is useful to assess the sensorimotor deficits underlying force control. Age-related impairments in grasping are also associated with decreased sensorimotor control [21,22]. Therefore, an analysis of the COP can provide important information to assess grasp control in older adults.

The aim of this study was to investigate the bimanual interference on finger spatial stability in precision grip for elderly people. In the experiment, we needed to address some technical challenges. Firstly, an analysis based on high spatial resolution was required to monitor the COP trajectory during holding an object. The problem was solved by a pressure sensor using a high spatial resolution with 248 sensors per cm^2^ [23]. The sensor sheets enabled accurate measurement of the COP trajectory and precise quantification of the spatial stability. Secondly, traditional methods using load cells strain gauges have a limitation in the assessment of grasp control of objects of different shapes and sizes. Therefore, we used a method that directly attached the sensor sheets to freely movable objects. Thin and flexible sensor sheets allowed for accurate assessment of the spatial stability during natural reach to grasp movements.

## 2. Materials and Methods

### 2.1. Subjects

We recruited twenty-one older adults (3 males and 18 females,) and twenty-one younger adults (3 males and 18 females). The average age was 78.5 (±5.6) years for the older adults and 22.0 (±2.7) years for the younger adults. The ranges of Edinburgh Handedness Inventory (EHI) were 67–100 (older adults) and −20–100 (younger adults). The inclusion criteria included the ability to manipulate objects bimanually with precision grip and to obey commands. The exclusion criteria included history of stroke and coexisting musculoskeletal, neurologic, or severe cognitive disorders. The study obtained informed consent from all subjects. The study was carried out in accordance with the Helsinki Declaration and was approved by the Ethical Review Board of the university with number #2016-110.

### 2.2. Experimental Procedures

The subjects were seated in a chair, and the height of the front table was adjusted. A cubic object made of iron (250 g, 31 × 31 × 31 mm) was on the table, 30 cm from the participants in the median sagittal plane. In order to decrease the difference between individuals in digital skin friction, the fingertips were wiped with alcohol-soaked cotton. The experiment consisted of a single task (ST) and a dual task (DT) session (Figure 1). In the ST, the experimental procedure was based on Westling and Johansson [12]. The instruction given to the participants were as follows: (1) grip the object with the thumb and index finger, (2) lift it at a height of about 5 cm, and (3) hold it for about 5 s. In the DT, a pegboard was placed on the table next to the cubic object and four pegs were used. The instructions were as follows: (1) grip the object with the thumb and index finger, (2) lift it at a height of about 5 cm, and (3) invert all four pegs using the other hand while holding the object. In both tasks, we required the participants to execute the task using minimum force. On the other hand, we did not ask the participants to focus on one task more than the other in the DT. Each session was repeated 10 times for the dominant and non-dominant hand, and an interval of 5 s was given. The testing order for the dominant and non-dominant hand was randomized.

### 2.3. Materials

Two pressure sensors (Pressure Mapping Sensor 5027, Tekscan, South Boston, MA, USA) were used. The sensors were mounted on the contact surface of the object. The detection area of the sensor was 27.9 × 27.9 mm. A total of 1936 sensing elements were distributed on the sensor sheet. Each element had a sensitivity range of 0–345 KPa, and the spatial resolution was 248 sensors per cm^2^. The sensor was thin (0.1 mm) and flexible enough to avoid interfering with the sensory aspect of gripping the object. The sensor worked as a variable resistor in the electrical circuit. The value of the resistance was changed depending on the force applied to the sensor. The resistance decreased when the force was applied and increased when the force was unloaded. The pressure distribution and magnitude were detected by reading the resistance value. One methodological challenge in using Tekscan sensors is the validation issue. Previous studies reported that Tekscan sensors are sensitive to surface condition, and to equilibration and calibration processes [24,25]. In order to respond to this challenge, we manufactured a flat-surfaces iron cube with a roughness (Ra) of 1.6 a. Moreover, equilibration and calibration were performed for each sensor sheet using dedicated equipment before each recording. I-scan 100 ver. 7.51 (Nitta, Osaka, Japan) was used to record the data of the pressure distribution, at 100 Hz frequency with a resolution of 8-bits.

GF control was evaluated using two kinetic parameters: the COP trajectory and the mean GF. The COP (mm/4 s) is the center of all applied forces and indicates the balance of the fingertip force on the sensor sheets. The COP coordinates in the *X-* and *Y*-axis were determined by the following equations:(1)$$Xcop=\frac{\begin{array}{cc} \begin{array}{c} \begin{array}{c} Cols-1\\ \sum^{\text{​}}\; \end{array}\\ i=0 \end{array} & \begin{array}{c} \begin{array}{c} Rows-1\\ \left(i^{*}\sum^{\text{​}}Fij\right) \end{array}\\ j=0 \end{array} \end{array}}{\begin{array}{cc} \begin{array}{c} \begin{array}{c} Cols-1\\ \sum^{\text{​}}\; \end{array}\\ i=0 \end{array} & \begin{array}{c} \begin{array}{c} Rows-1\\ \left(i^{*}\sum^{\text{​}}Fij\right) \end{array}\\ j=0 \end{array} \end{array}}\;$$
and
(2)$$Ycop=\frac{\begin{array}{cc} \begin{array}{c} \begin{array}{c} Rows-1\\ \sum^{\text{​}}\; \end{array}\\ i=0 \end{array} & \begin{array}{c} \begin{array}{c} Cols-1\\ \left(i^{*}\sum^{\text{​}}Fij\right) \end{array}\\ j=0 \end{array} \end{array}}{\begin{array}{cc} \begin{array}{c} \begin{array}{c} Rows-1\\ \sum^{\text{​}}\; \end{array}\\ i=0 \end{array} & \begin{array}{c} \begin{array}{c} Cols-1\\ \left(i^{*}\sum^{\text{​}}Fij\right) \end{array}\\ j=0 \end{array} \end{array}},$$
where *F* is the force at each sensel and the calculation of the COP takes all forces on the sensor into account [26]. The COP displacement on the senser sheets was recorded every 0.01 s. The recording data was converted to a spreadsheet. In the spreadsheet, the location of the COP on the *X-* and *Y*-axis (mm) was displayed as a digital value. The distance of the COP displacement (*COP_d_*) in a two-dimensional surface in 0.01 s was calculated by the following equation:(3)$$COP_{d}=\sqrt{{\left(Xcop_{2}-Xcop_{1}\right)}^{2}+{\left(Ycop_{2}-Ycop_{1}\right)}^{2}}.\;$$

The total length of the COP for the first 4 s from the start of the lift was calculated. Another parameter was the mean GF (N/4 s), indicating the amplitude of forces. The sum of the GF applied to the sensor sheets for the first 4 s was calculated. In addition, a change in DT performance in the COP trajectory and mean GF was evaluated as dual task costs (*DTC*). The parameter indicates the influence in performance from *ST* to *DT*. The *DTC* was calculated by the following equation:(4)Dual task costs (DTC)=DT−ST.

### 2.4. Clinical Evaluation

Trail Making Tests A and B (*TMT-A* and *-B*) were used to evaluate cognitive processing speed and executive attention function. The *TMT-A* is associated with selective attention and the *TMT-B* adds a measure of attentional allocation. In addition, the level of change in the completion time from the *TMT-A* to the *TMT-B* was measured as Δ*Trail Making Test* (Δ*TMT*). The Δ*TMT* is a more accurate evaluation of executive function and removes the effects of upper extremity function. The Δ*TMT* was calculated by the following equation:(5)Δ Trail Making Test (ΔTMT)=TMT−B−TMT−A.

### 2.5. Statistical Analysis

The Shapiro–Wilk test was used to evaluate the normal distribution of data. As a result, the normal distribution was confirmed for the TMT-A and -B in older adults. In younger adults, normal distribution was confirmed only in the TMT-A. Differences between older and young adults were tested using the Chi-square test for sex ratio, the unpaired t test for the TMT-A, and the Mann–Whitney test for the TMT-B and ΔTMT.

A three-way ANCOVA with ΔTMT as a covariate was conducted for the length of COP trajectory and the GF. The factors were ‘age’ (older adults and younger adults), ‘task difficulty’ (ST and DT), and ‘hand’ (dominant hand and non-dominant hand).

The correlation coefficient was used to measure the association between the DTC of kinetic parameters and ΔTMT for older and younger adults, respectively. The analyses were conducted based on the software SPSS version 23.0 J (IBM SPSS Japan., Tokyo, Japan). As the effect size, the correlation coefficient (r) was used for the unpaired test, Chi-square test, and Mann–Whitney test. Moreover, partial eta squared (*η**_p_*^2^) was used for the ANCOVA. The significance level was set at *p* < 0.05.

## 3. Results

Table 1 demonstrates the comparison results between the older and younger adults. The ranges of age were 70–89 (older adults) and 21–31 (younger adults). The TMT-A, TMT-B, and ΔTMT were significantly higher in the older adults.

Figure 2 demonstrates a typical example of the COP trajectories (A) and the GF traces (B) for younger and older adults. Table 2 summarizes the mean value of the kinetic parameters and DTC.

Table 3 demonstrates a comparison of the kinetic parameters. In the COP trajectory, the main effects of age for the thumb (*F*_1,__140_ = 46.17, *p* < 0.01) and index finger (*F*_1,140_ = 22.14, *p* < 0.01) and task difficulty for the thumb (*F*_1,140_ = 6.47, *p* = 0.01) were significant. There was a significant interaction between age and task difficulty for the thumb (*F*_1,140_ = 4.57, *p* = 0.03). In the GF, the main effects of age for the thumb (F_1,140_ = 96.58, *p* < 0.01) and index finger (*F*_1,140_ = 107.72, *p* < 0.01), task difficulty for the thumb (*F*_1,140_ = 11.76, *p* < 0.01) and index finger (*F*_1,140_ = 8.20, *p* < 0.01), and hand in the index finger (*F*_1,140_ = 7.66, *p* < 0.01) were significant. There was a significant interaction between age and hand for the index finger (*F*_1,140_ = 8.77, *p* < 0.01).

Table 4 demonstrates correlations between the DTC of kinetic parameters and the ΔTMT based on Pearson’s and Spearman’s rank correlations. In the older adults, the DTC in the GF for the non-dominant thumb and index finger were correlated positively with the ΔTMT. In younger adults, the DTC in the COP trajectories for the dominant thumb was correlated negatively with the ΔTMT. The DTC in the GF for the dominant thumb and index finger were correlated positively with the ΔTMT.

## 4. Discussion

The COP trajectory within the fingertip was significantly longer in the younger adults and the GF was larger in the elderly adults. Based on our previous study investigating the COP trajectory in the paretic finger, we expected that the COP trajectory will be increased in the elderly adults because the decreased sensorimotor control in the elderly could result in an increased COP trajectory [16]. However, contrary to our expectations, the COP trajectory was longer in the younger adults than that of the elderly adults. The result may indicate that the age-related changes of the COP trajectory occur by a different mechanism than pathological changes in stroke.

The COP trajectory was longer in the younger adults. This suggests a difference in grasping strategies between the younger and older subjects. In other words, younger adults strategically prioritized energy conservation at the expense of finger spatial stability, and older adults used a strategy that prioritized spatial stability over energy conservation. Similar strategies have found in postural control. Several studies have shown that the COP displacement is reduced in standing postural control on high places [27,28]. This is a strategy to compensate for postural stability in order to prevent falls based on negative psychological states such as fear of heights or falls. Furthermore, Brown et al. found that the elderly prioritized postural stability with reduced COP area at the expense of performance of the secondary task in conditions involving increased postural threat [29]. Considering the strategic difference in the context of precision grip, the result in this study showed that the elderly compensated for finger spatial stability by disregarding grasping efficiency in order to prevent the object slipping. This hypothesis is supported by previous reports demonstrating that younger adults use minimum GF as an energy conservation strategy, while older adults use twice as much GF as younger people [21,30]. Actually, the mean GF value (older: 2618 ± 1024 (N), younger: 1021 ± 532 (N)) and COP trajectory (older: 16.5 ± 4.8 (mm), younger: 29.8 ± 10.8 (mm)) within the dominant thumb in the ST demonstrated obvious age-related differences. Our findings provide new insights into age-related differences in grasping strategies in terms of finger spatial stability.

One of the key findings in this study was that the trajectory of the COP decreased, while the GF increased in the dual motor task in both younger and older groups. This means that the safety strategies responding to dual task interference are universal regardless of age. The effect of dual task interference on a precision grip task has been described primarily in temporal and dynamical parameters. Regarding the temporal parameters, previous studies have reported that the preload phase (time between first contact of the fingertips with the object and onset of the load force) or the lift phase (time for the object to be lifted and stabilized) increased in the dual task [31,32]. Moreover, in the dynamical parameters, increased maximum GF or mean GF in the hold phase in the dual task were reported [31,32,33]. Changes in these parameters may be due to sensorimotor processing for precise grasp control being interfered with by increased information load. In contrast to the numerous studies adopting motor-cognitive dual tasks, studies with dual motor tasks, especially focusing on bimanual interference on precision grip, are limited. It is only known that performing a lifting or tapping task with one hand increased the GF of the other hand during object lifting [10]. Competition for processing resources between tasks may have prevented proper allocation to the precision grip-lift task, resulting in an increased safety margin to prevent the object slipping. In the present study, we observed increased GF in the dual task, and this is consistent with previous studies. On the other hand, it is the first to describe changes in the finger COP trajectory. The results suggest that the safety strategy is even applied to increased processing load, in addition to decreased sensorimotor control with aging.

We also found that the GF in the non-dominant index finger was smaller than that of the dominant hand. This suggests that the non-dominant hand has an advantage in terms of static force control while holding objects. This finding may support previous studies demonstrating functional differences between the dominant and non-dominant arm. Specifically, the dominant arm is specialized for dynamical control, but is susceptible to perturbations, whereas the non-dominant one is specialized for position control and less prone to perturbations [34,35]. Because the object lifting task in this study was essentially static, the functional advantage in the non-dominant hand may have been reflected in the efficient GF control. Indeed, Ferrand et al. found a smaller GF/lord force ratio in the non-dominant hand in static bimanual manipulation tasks [36]. This means that the static force regulation is more economical in the non-dominant hand. Moreover, the small GF in the non-dominant hand may have occurred because the object lifting task was performed under an asymmetric bimanual condition. If the pegboard manipulation with one hand had indirectly acted as a perturbation, the GF control may have been optimized by the advantage of the non-dominant hand.

The DTC of the kinetic parameters was significantly associated with the ΔTMT. In this study, the ΔTMT was calculated from the time difference between the TMT-A and -B and used to evaluate the executive function more accurately. One notable finding was that the DTC of the GF showed a significant positive correlation with the ΔTMT in both the older and younger adults. This result suggests that the subjects with lower executive function exerted the larger GF in the dual task condition regardless of age. In other words, this means that higher cognitive processes are required for the GF control in bimanual manipulation. The finding may be supported by previous reports explaining the relationship between bimanual coordination and executive function. Bangert et al. demonstrated that temporal coupling for limbs in two-handed circle-drawing and tapping tasks was related to Backward Digit Span and self-reported executive control [37]. Furthermore, Kim et al. found that synchronization errors in between-hand rhythmic movements were related to the TMT and Digit Span Test [38]. These previous findings indicated that executive function played an essential role in temporal control in bimanual movements. In addition, the results of this study further suggested that executive function is just as important in precise force control. As another possibility, simultaneous performance of the pegboard task may have led to increased GF. Several studies reported that executive function was linked with pegboard task performance [39,40,41]. Therefore, it is possible that the interference with the GF control was increased in subjects with poor executive function because they needed to allocate more resources to the pegboard task.

However, interestingly, the hand associated with the ΔTMT was different between younger and older subjects. In the older subjects, the DTC of the GF was related to the executive function only in the non-dominant hand. Previous studies showed that bimanual task performance was associated with the executive function in elderly people only in the non-dominant hand [41,42]. These findings are consistent with the results in this study, suggesting that the non-dominant hand in the older adults may be supported by higher cognitive processing. According to Hibino & Gornial, the non-dominant hand in older adults was more involved in stabilizing the object when an unexpected perturbation occurred in the dominant one during a bimanual grasping task [43]. Furthermore, Noble et al. showed that the GF control in elderly people required higher activation in the areas related to executive processing [44]. Based on these findings, the advantage of the non-dominant hand and the stability of the precision grip may be reinforced by execution control.

In contrast to the older subjects, the DTC of the GF was only associated with the executive function in the dominant hand in the younger adults. Janssen et al. revealed that the end-state comfort effect, an indicator of predictive motor planning, was found only in younger subjects’ right hands during bimanual manipulation [45,46]. Although the direct relationship is not known, our speculation is that aspects of motor planning were reflected in the dominant hand, even in the precision grip task, and the relationship with executive function became apparent. According to Kalisch et al., the superiority of the dominant hand gradually decreases with age [47]. However, it is not clear how such changes are associated with executive function and how they occur in the process of aging. The results provide a clue that executive control may affect the age-related difference in grasping strategies. We need further experimental studies focusing on higher cognitive aspects, such as measuring real-time changes in cognitive processing while not restricting natural bimanual grasping.

## 5. Limitations

There are several methodological limitations. In the dual task, we could not identify which phase during the process of gripping and lifting the object was interfered with by the peg board task. This was because there were not any techniques to synchronize the kinetic parameters with the performance of the peg board in our laboratory. In order to respond to this limitation, the first 4 s, which includes the time for manipulation of pegs, were assessed in this study. In future research, a novel technique synchronizing two motor tasks may provide useful information regarding the dual task interference in the process of the GF control. Moreover, the analysis of the COP displacement targeted only the length of the trajectory. Novel techniques, such as quantifying the range or direction of the COP, will lead to analysis of different aspects that capture how the spatial stability is compensated. Lastly, in the dual task, attentional priority was not considered. In a previous study, finger force control was affected by the task prioritization in the posture-motor dual task [48]. Future work may also clarify the effects of task prioritization on the finger spatial stability in the bimanual task.

## 6. Conclusions

In the present study, finger spatial stability during precision grip was compared between older and younger adults, and between single and bimanual manipulation (Section 2.2). Two kinetic parameters of the COP trajectory and the GF calculated from a Teksacn pressure sensor system were used (Section 2.3). As results, the COP trajectory was decreased in the older adults. Furthermore, decreased COP displacement was also observed in bimanual manipulation regardless of age (Table 3). These results suggest a safety strategy to prioritize spatial stability is applied to both the decreased sensorimotor control with aging, and the increased processing load with the dual motor task. This study provides new insights into the interpretation of the COP and contributes to the further potential of sensor devices in the fields of health science and rehabilitation.

## Figures and Tables

**Figure 1 sensors-21-08396-f001:**
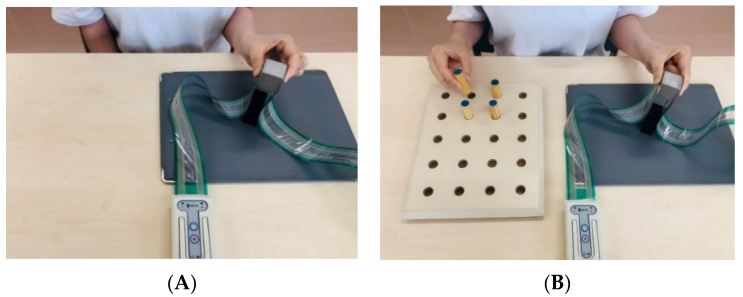
Grip-lifting task: (**A**) Single task; (**B**) Dual task.

**Figure 2 sensors-21-08396-f002:**
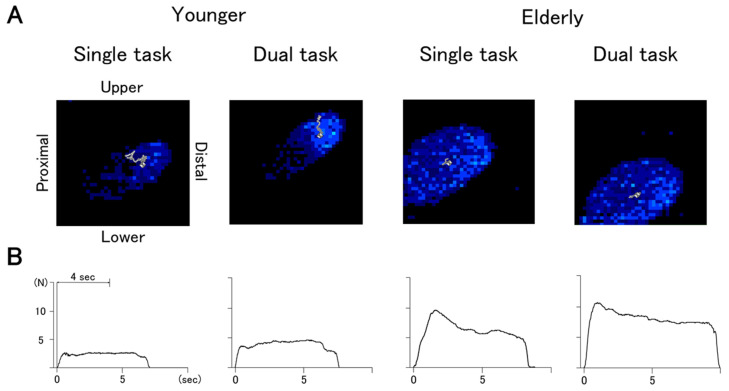
The trajectories of the center of pressure (**A**) and the traces of grip force (**B**) for typical subjects. The contact area of the fingertips is shown in blue, and the trajectory of the COP is shown by a gray line in (**A**).

**Table 1 sensors-21-08396-t001:** Comparison between the older and younger adults.

	Older (n = 21)	Younger (n = 21)	*p*	*r*
Age	78.5 ± 5.6	22.0 ± 2.7	<0.01	0.88
Sex (M/F)	3/18	3/18	0.67	<0.01
Trail Making Test A (s)	134.6 ± 57.4	65.6 ± 13.7	<0.01	0.92
B (s)	164.8 ± 56.4	69.9 ± 22.4	<0.01	0.79
Δ Trail Making Test (s)	47.6 ± 45.1	4.3 ± 18.6	<0.01	0.58

**Table 2 sensors-21-08396-t002:** Mean value of kinetic parameters and dual task costs.

		COP Trajectory			Mean Grip Force		
		ST (mm) ^a^	DT (mm) ^b^	DTC (mm) ^c^	ST (N) ^a^	DT (N) ^b^	DTC (N) ^c^
Older							
Dominant	Thumb	16.5 ± 4.8	14.8 ± 3.5	−1.1 ± 3.2	2618 ± 1024	3310 ± 1222	577 ± 827
	Index finger	15.5 ± 5.4	15.6 ± 4.9	0.6 ± 3.7	3348 ± 1107	3922 ± 1194	438 ± 795
Non-	Thumb	15.1 ± 4.3	15.3 ± 4.6	0.3 ± 2.9	3157 ± 1021	3569 ± 1008	333 ± 597
	Index finger	17.6 ± 5.9	18.2 ± 5.4	1.1 ± 5.0	2695 ± 988	3011 ± 1153	258 ± 695
Younger							
Dominant	Thumb	29.8 ± 10.8	22.7 ± 10.2	−7.1 ± 8.3	1021 ± 532	1581 ± 705	561 ± 490
	Index finger	28.7 ± 14.2	22.7 ± 11.9	−6.0 ± 7.8	999 ± 553	1516 ± 722	517 ± 490
Non-	Thumb	28.8 ± 9.1	23.7 ± 9.0	−5.1 ± 8.8	962 ± 433	1321 ± 1153	359 ± 430
	Index finger	24.8 ± 8.8	20.8 ± 7.4	−4.0 ± 7.5	1095 ± 459	1475 ± 607	380 ± 497

^a^ Single task, ^b^ Dual task, ^c^ Dual task costs.

**Table 3 sensors-21-08396-t003:** Results of three-way ANCOVA for the effect of age, task, and hand on kinetic parameters.

	Source	Sum of Squares	df	*F* Value	*p* Value	*η_p_* ^2^
COP trajectory (mm)						
Thumb	Age	2876.16	1	46.17	<0.01	0.25
	Task	403.04	1	6.47	0.01	0.04
	Hand	1.46	1	0.02	0.87	<0.01
	Age × Task	284.78	1	4.57	0.03	0.03
	Age × Hand	1.00	1	0.02	0.89	<0.01
	Task × Hand	23.67	1	0.38	0.53	<0.01
	Age × Task × Hand	1.24	1	0.02	0.88	<0.01
Index finger	Age	1698.597	1	22.14	<0.01	0.14
	Task	185.656	1	2.42	0.12	0.02
	Hand	1.00	1	0.01	0.90	<0.01
	Age × Task	272.13	1	3.54	0.06	0.03
	Age × Hand	263.786	1	3.44	0.06	0.02
	Task × Hand	5.530	1	0.07	0.78	<0.01
	Age × Task × Hand	12.67	1	0.17	0.68	<0.01
Mean grip force (N)						
Thumb	Age	69,304,377.38	1	96.58	<0.01	0.41
	Task	8,438,874.61	1	11.76	<0.01	0.08
	Hand	288,028.74	1	0.40	0.52	<0.01
	Age × Task	15,099.02	1	0.02	0.88	<0.01
	Age × Hand	2,253,441.26	1	3.14	0.08	0.02
	Task × Hand	364,154.87	1	0.50	0.47	<0.01
	Age × Task × Hand	34.48	1	<0.01	0.99	<0.01
Index finger	Age	81,822,564.45	1	107.72	<0.01	0.44
	Task	6,228,928.25	1	8.20	<0.01	0.06
	Hand	5,819,038.81	1	7.66	<0.01	0.05
	Age × Task	47,802.17	1	0.06	0.80	<0.01
	Age × Hand	6,658,118.63	1	8.77	<0.01	0.06
	Task × Hand	207,754.248	1	0.27	0.60	<0.01
	Age × Task × Hand	1792.31	1	<0.01	0.96	<0.01

**Table 4 sensors-21-08396-t004:** Correlations between the DTC of kinetic parameters and ΔTMT.

			ΔTMT ^a^	
			Older ^b^	Younger ^c^
DTC				
COP trajectory	Dominant	Thumb	−0.22	−0.46 *
		Index finger	−0.06	−0.37
	Non-	Thumb	0.17	0.05
		Index finger	−0.32	0.04
Mean grip force	Dominant	Thumb	−0.26	0.56 **
		Index finger	−0.21	0.50 *
	Non-	Thumb	0.58 *	−0.02
		Index finger	0.55 *	0.04

* *p* < 0.05; ** *p* < 0.01, ^a^ ΔTrail Making Test, ^b^ Pearson’s correlation, ^c^ Spearman’s rank correlation.

## Data Availability

The data presented in this study are available on request from the corresponding author. The data are not publicly available due to ethical reasons.

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
