# Peer review of "Aging and Bimanual Effects on Finger Center of Pressure during Precision Grip: Different Strategies for Spatial Stability"

_sensors, 2021, doi:10.3390/s21248396_

Round 1

Reviewer 1 Report

The work reveals that during the precision grip, the younger prioritize energy conservation strategy while the elder prefer safety strategy with larger grip force and decreased COP displacement.Furthermore, the research also shows that the safety strategy is adapt to bi-manual manipulation regardless of age.As the research results are based on the pressure sensors, the work shows the potential of sensor devices in the fields of health science and points out the insufficient of current sensors,which may guiding the future development of the sensor devices.However,there are still some flaws which can be improved.

  1. The English expression needs to be improved and some spelling errors need to be corrected, such as line 18, line 28 and line 118, please check the whole paper carefully.
  2. Some important information is not clearly expressed, such as the precise definition of COP and the process to get the COP from the pressure sensors, please give detailed supplementary explanation.
  3. The conclusions are not intuitive to describe only in words, maybe highlight the key indicators on the Tables can be better.
  4. The formula expression in this paper is not rigorous enough, please modify the expression.

Reviewer 2 Report

 The paper examines the bimanual interference on finger spatial
stability during precision grip in older adults. 

Title:

Maybe it needs to be revised to describe the comparison in grasping strategies between younger and older people.

Abstract:

It describes clearly the goal of the paper.

Introduction

It includes the importance of research and surveying related works.

Materials and methods

It is abstracted and it needs expansion and more details.

There are no equations or theory governing the topic.

More details about the sensor and its circuit and specifications are needed.

Why 21 persons with the given configuration of male/female structure.

Results

It is detailed.

Discussion

The authors list some findings which is good. However, I wonder, if those findings are novel or new or do, they conflict with previous works or in harmony with it?

Conclusion

It is clear and informative though there is a room for more improvement by adding briefly some of the findings (which we asked about) if they are novel.

Reviewer 3 Report

The manuscript reported on age-related differences in finger spatial stability during a single and a dual task. The study seems interesting in some regards, yet I find that the overall goal is not so clear and the presentation and organization of the manuscript need further improvement. Here are some major concerns.
- The title is not informative
- The abstract does not provide sufficient information on the study and study results.
- In the introduction, it was noted "we focused on a method that directly attached the sensor sheets to freely movable objects". I understand that the authors used Tekscan sensors, yet it is difficult to see why this was a methodologically challenging and what the authors did to overcome the challenge. This may be because I've seen using Tekscan over various object surfaces. Validation (of Tekscan measures) may be challenging in some cases.
- For the dual task, it might be informative to report the time taken to invert pegs for young and older participants. 
- Clinical evaluation was included in the methods section, but its results were not well integrated throughout the manuscript.   
- In Statistical analysis, it is unclear what "Chi-square test (sex)" means. Were gender effects considered? Did the authors explore three-way interaction effects and not include them since they were not significant? I wonder whether ΔTMT should have been explored as a covariate.
- Figure 5. It is not so clear why the first 4-second data were examined. At least, it may be more informative to separate the initial transition phase (i.e., the part the force increases) and the steady state phase. However, if there is a justification for the 4-sec duration, please provide.
- This discussion doesn't seem to reflect the study focuses stated by the authors. 

Round 2

Reviewer 2 Report

The authors responded to the comments given to them and did the required modifications.

Reviewer 3 Report

I thank the authors for the revised version. I don't have any major concern, yet have three requests (the latter one is a moderate concern)

  • The title, Aging and bimanual effects on finger center of pressure: Different strategies for spatial stability may be: Aging and bimanual effects on finger center of pressure during precision grip: Different strategies for spatial stability. This is for clarity. 
  • Equations 1-4 don't look right on my mac. The authors may need to check the format again.
  • The authors did not include the instruction provided to participants for the dual task. Were they asked to focus more on one task than the other (i.e., was a primary task defined in the dual task?). Please indicate in the methods, and discuss how that might have affected results.
